# Alloreactive Immune Response Associated to Human Mesenchymal Stromal Cells Treatment: A Systematic Review

**DOI:** 10.3390/jcm10132991

**Published:** 2021-07-05

**Authors:** Raquel Sanabria-de la Torre, María I. Quiñones-Vico, Ana Fernández-González, Manuel Sánchez-Díaz, Trinidad Montero-Vílchez, Álvaro Sierra-Sánchez, Salvador Arias-Santiago

**Affiliations:** 1Cell Production and Tissue Engineering Unit, Virgen de las Nieves University Hospital, Andalusian Network of Design and Translation of Advanced Therapies, 18014 Granada, Spain; raquelsanabriadlt@gmail.com (R.S.-d.l.T.); maribelqv@ugr.es (M.I.Q.-V.); alvarosisan@gmail.com (Á.S.-S.); salvadorarias@ugr.es (S.A.-S.); 2Biosanitary Institute of Granada (ibs.GRANADA), 18012 Granada, Spain; manolo_94_sanchez@hotmail.com (M.S.-D.); tmonterov@gmail.com (T.M.-V.); 3Department of Dermatology, Virgen de las Nieves University Hospital, 18014 Granada, Spain; 4Department of Dermatology, Faculty of Medicine, University of Granada, 18071 Granada, Spain

**Keywords:** adverse events, alloantibodies, clinical trials, donor specific antibodies, immunogenicity, mesenchymal stromal cells, safety

## Abstract

The well-known immunomodulatory and regenerative properties of mesenchymal stromal cells (MSCs) are the reason why they are being used for the treatment of many diseases. Because they are considered hypoimmunogenic, MSCs treatments are performed without considering histocompatibility barriers and without anticipating possible immune rejections. However, recent preclinical studies describe the generation of alloantibodies and the immune rejection of MSCs. This has led to an increasing number of clinical trials evaluating the immunological profile of patients after treatment with MSCs. The objective of this systematic review was to evaluate the generation of donor specific antibodies (DSA) after allogeneic MSC (allo-MSC) therapy and the impact on safety or tolerability. Data from 555 patients were included in the systematic review, 356 were treated with allo-MSC and the rest were treated with placebo or control drugs. A mean of 11.51% of allo-MSC-treated patients developed DSA. Specifically, 14.95% of these patients developed DSA and 6.33% of them developed cPRA. Neither the production of DSA after treatment nor the presence of DSA at baseline (presensitization) were correlated with safety and/or tolerability of the treatment. The number of doses administrated and human leucocyte antigen (HLA) mismatches between donor and recipient did not affect the production of DSA. The safety of allo-MSC therapy has been proved in all the studies and the generation of alloantibodies might not have clinical relevance. However, there are very few studies in the area. More studies with adequate designs are needed to confirm these results.

## 1. Introduction

In 2006 the International Society for Cell and Gene Therapy (ISCT) set up minimal criteria defining mesenchymal stromal cells (MSCs). Firstly, MSCs must be plastic-adherent when maintained in standard culture conditions. Secondly, they must express CD73, CD90 and CD105 and lack expression of hematopoietic and endothelial markers CD11b, CD14, CD19, CD34, CD45, CD79alpha and HLA-DR. Finally, MSC must be capable of in vitro differentiation into adipocyte, chondrocyte and osteoblast lineages [1,2].

Among the various sources of MSCs, perinatal tissues have special interest. Tissues associated with birth, such as the placenta, umbilical cord blood, amniotic fluid and amnion, are widely available and can be used for therapeutic purposes [3]. Adult MSCs can be obtained from different tissues although the most widely used are bone marrow, peripheral blood or adipose tissue of patients [4]. The ease of isolation, cultivation, and the high ex vivo expansion potential in line with the numerous therapeutic mechanisms (paracrine pro-regenerative, anti-fibrotic, antiapoptotic, pro-angiogenic and immunomodulatory functions) have contributed to this broad exploitation (Figure 1) [5].

MSCs treatment can be autologous, allogeneic and xenogeneic. Xenogeneic MSCs treatment is often unsuccessful due to irreconcilable inter-species differences. Auto-MSCs treatment involves donor site morbidity. Age and disease state of the patient can also adversely affect the quality of the cells [6]. Furthermore, expanding these cells to a sufficient number for therapy is time-consuming. Therefore, allo-MSCs represent one of the most promising candidates for regenerative medicine applications [6]. The use of allo-MSC has the advantage of having low immunogenicity [3]. Cells can be obtained from healthy donors followed by in vitro expansion, cryopreservation storage, and subsequent transport when necessary (Figure 1).

Furthermore, MSCs have been considered “immunoprivileged”, as they do not activate aggressive immune responses. Numerous in vitro and in vivo studies have shown that MSC therapy promotes angiogenesis and the growth and differentiation of local progenitor cells. MSC therapy prevents fibrosis and apoptosis, attracts immune cells to the site of injury and modulates immune responses. There is current evidence showing that MSCs achieve a therapeutic effect in vivo not due to their proliferative and immunomodulatory capacity but also through paracrine signaling [4,7,8]. In 2018, the first marketing authorization for an MSC product was granted by the European Medicines Agency (EMA) for the treatment of complex perianal fistulas in patients with Crohn’s disease (CD). From a regulatory perspective, MSCs are classified as an advanced therapy medicinal product (ATMP) [9,10]. The hypoimmunity of MSCs is due to the lack of expression of Major Histocompatibility Complex II (MHC II), as well as the low expression of MHC I and costimulatory molecules such as CD80 and CD86 [11]. Thus, MSCs are not detected by effector CD4+ T cells. Furthermore, MSCs negatively regulate the Fas ligand/Fas receptor on the cell surface, keeping T cells in a dormant state. Through IL-10, MSCs favor the passage of macrophages of pro-inflammatory phenotype M1 to anti-inflammatory M2. MSCs secrete the HLA-G-5 and HLA-G-7 isoforms, which keep immune tolerance under control, having a powerful influence on allograft tolerance [3].

On the other hand, MSCs secrete paracrine factors, including growth factors, exosomes, cytokines and chemokines, which promote the formation of a regenerative environment by inhibiting B and T cell proliferation and monocyte maturation. MSCs also stimulate the generation of regulatory T cells and M2 macrophages [12,13,14]. So, MSCs are considered safe for use in allogeneic settings without concern for immune rejection. The exosomes secreted by MSCs carry DNA, RNA, miRNA, proteins and other factors that act as messengers in intercellular communication [4].

Although the majority of in vitro studies have highlighted the immunosuppressive properties of MSCs, several studies have provided evidence that mismatched MSCs are immunogenic [15]. Mismatches in HLA antigens between donor and recipient are a strong immune barrier that leads to serious complications such as graft failure, transplant rejection, or graft versus host disease (GVHD) [3]. When MSCs are exposed to a pro-inflammatory environment or during the MSCs differentiation process, these cells increase their surface immunogenicity. Regarding preclinical studies, memory T lymphocytes and alloantibodies have been detected in mice. MSCs cannot completely evade the immune system and are eventually rejected [16]. Nauta et al. demonstrated that MSCs induce a memory T-cell response which results in donor graft rejection in a nonmyeloablative setting [17]. The presence of lytic alloantibodies after inoculating allo-MSCs has been reported in pigs [18], rats [19] and baboons [20]. Cho et al. showed significant increases in anti-donor immunoglobulin G titers 7 days after subcutaneous administration of IFNγ-activated allogeneic MSCs (allo-MSCs) in pigs [21]. Camp et al. observed increased numbers of neutrophils, monocytes and T cells at the site of intracranial injection of allo-MSCs in rats [22]. Joswig et al. observed an increase in total nuclear synovial cells in horse after a second injection of allo-MSCs, a fact that did not occur with the same treatment carried out with autologous MSCs (auto-MSCs) [23]. Berglund et al. reported that the antisera from all 4 horses injected with MHC-mismatched MSCs contained antibodies causing the death of leukocyte antigen A2 (ELA-A2) haplotype MSCs in the microcytotoxicity assays [24]. Isakova et al. analyzed the immune response in rhesus macaques after intracranial administration of allo versus auto-MSCs, showing an allorecognition based on the expansion of subsets of Natural Killer (NK), B and T cells in peripheral blood and the detection of allogeneic antibodies in recipients of allogeneic transplants [25]. Furthermore, the magnitude of the response was influenced by the degree of mismatch between the donor and the recipient of MSCs. An inverse correlation was also established between cell dose and MSCs engraftment levels in the brain six months after treatment [25]. Repeated injection of allo-MSCs in horses resulted in primary and secondary humoral responses. The immunomodulatory profile of MSCs prepared with pro-inflammatory cytokines could have helped them more effectively regulate inflammation and elicit a lower primary humoral response when first administered. However, if injected repeatedly, pre-existing antibodies could be more easily targeted to MSCs due to their higher expression of MHC. A partial match between donor and recipient may help avoid this secondary humoral response after re-exposure to MSC [26]. Even though most MSCs are used for their paracrine signaling effects rather than for expected differentiation or engraftment into host tissue, they still need to persist throughout the inflammatory phase and into the remodeling phase for maximal therapeutic benefit [24]. However, there is evidence that cell-mediated alloimmune responses can limit the persistence of MSCs in vivo [27]. In the study by Berglund et al. [26], all horses administered non-stimulated HLA-mismatched allo-MSCs developed de novo antibodies on day 14. These cell therapy induced alloantibodies contributed to the targeted killing of MSCs in vivo. Repeated injections of MSC would result in accelerated rejection of the cells, further limiting its beneficial effects and increasing the potential for adverse events. Injection of donor MSCs into less vascularized tissues like tendons and joints may result in different responses than intradermal or intravascular administration, during injury even tendons and joints are infiltrated with immune cells [28] that can contribute to allorecognition and subsequent immune rejection of MSCs. As the therapeutic benefits of MSCs appear to be largely due to the secretion of paracrine factors that promote healing of healthy tissue [29], it is necessary for the cells to persist throughout the initial inflammatory and healing period. Targeted destruction of allogeneic, MHC mismatched MSCs shortly after transplant would therefore limit their therapeutic potential.

Regarding clinical translation of MSCs, MSCs therapies are effective in many diseases such as GVHD [30,31], ischemic stroke [32], tuberculosis [33], osteoarthritis [34], kidney disease [35], hyperglycemia [36] and human immunodeficiency virus (HIV) infection [37]. There are currently more than 1200 clinical trials investigating the role of MSCs in musculoskeletal, neurological, traumatic, dermatological and autoimmune diseases [38,39]. Recently, the role of MSCs therapy in acute respiratory distress syndrome (ARDS) due to severe cases of COVID-19 infection has been studied. In these patients, MSCs have demonstrated safety and possible efficacy [40] because of their anti-inflammatory, antiapoptotic, anti-microbial and pro-angiogenic effects [41,42]. MSCs also promote bacterial and alveolar fluid clearance and disrupt the pulmonary endothelial and epithelial cell damage as they are capable of transferring mitochondria to injured epithelial cell [40]. Furthermore, angiopoietin-1 and keratinocyte growth factor secretion contribute to the restoration of alveolar–capillary barriers [43]. Regarding clinical trials, improved radiographic findings, pulmonary function and inflammatory biomarker levels have been found [41].

Recent data indicate that MSC treatment may provoke donors’ humoral and cellular immune responses, especially in allogeneic transplants. Detection of donor specific antibodies (DSA) in the serum of transplant recipients provides clear evidence of alloantigen recognition by B cells [44]. The generation of DSA is likely the results of indirect recognition of MSC HLA presented by patient antigen presenting cells (APC) to CD4+ T cells. As a result, induction of allo-specific T CD4+ cells will activate HLA-specific IgG-producing B cells.

At present, few studies address aspects of the safety or tolerability of allo-MSCs therapy to their immunogenicity. This systematic review aims to analyze the studies that evaluate the alloantibodies’ development in humans, including level of evidence, in order to estimate if the production of DSA is relatively common in the allo-MSC therapy, and if this alloantibodies’ production influences the safety or tolerability of the treatment.

## 2. Materials and Methods

### 2.1. Research Questions

−What is the prevalence of alloantibodies’ development in patients treated with allo-MSCs?−Does the route of administration of allo-MSCs influence the development of alloantibodies?−Does the tissue of origin of the MSCs influence the development of alloantibodies?−Is the total dose of allo-MSCs associated with increased alloantibodies response?−Are repeated doses of treatment associated with higher frequency of alloantibodies or impact of treatment?−Does the presensitization status of patients influence the development of alloantibodies?−How relevant is alloantibodies’ development to the safety of the treatment?

### 2.2. Search Strategy

This review was carried out in accordance with the Preferred Reporting Items for Systematic Reviews and Meta-Analyses: The PRISMA statement [45]. A bibliographic search was performed on May 10, 2021, using four different databases: Cochrane, Pubmed, MEDLINE and Scopus. The search algorithm used was “ALLOGENEIC MESENCHYMAL STROMAL CELL” OR “ALLOGENEIC MESENCHYMAL STEM CELL” AND “ANTIBODIES” OR “ALLOANTIBODIES”.

### 2.3. Eligibility and Exclusion Criteria

Randomized clinical trials which evaluated the production of DSA after allo-MSC treatment were selected. The search was focused on studies published in English from conception on 10 May 2021. The exclusion criteria were review articles, preclinical research articles and other types of publications that were not clinical trials, non-randomized clinical trials and case reports. Stem cell sources other than mesenchymal stem cells were also excluded. Finally, a total of 13 randomized clinical trials in which DSA were measured after treatment with allo-MSCs were obtained.

### 2.4. Study Selection

To carry out the study selection, a two-step review was applied. The titles and abstracts were screened independently by two authors (RST and MQV), and all publications reporting the measurement of DSA in the allo-MSC treated patients were included. Secondly, the entire publication was reviewed to ensure that it fulfilled the rest inclusion criteria. Disagreements about inclusion or exclusion of articles were discussed until a consensus was reached. If not reached, resolution was achieved by discussion with a third independent author (AFG).

### 2.5. Variables

The variables assessed were study design, follow-up, author, country, age, disease, number of participants, controls, cell dose, number of injections, route of delivery and MSC origin. Safety and tolerability were also analyzed as well as the efficacy of MSC treatment. Severe adverse events were divided into two groups: adverse events general group and severe adverse events. Regarding efficacy, the main endpoints and the results obtained after therapy are highlighted. Finally, a correlation is established between the development of DSA and the safety of the treatment.

### 2.6. Risk of Bias Assessment

The quality of the design was critically appraised using the Cochrane risk of bias assessment tool. This tool includes 7 specific domains: random sequence generation, allocation concealment, blinding of participants and personnel, blinding of outcome assessment, incomplete outcome data, selective reporting and other sources of bias. Studies were classified as having a low, high or unclear risk of bias. Any discrepancies between reviewers were resolved in consultation with the third author until a consensus was reached.

## 3. Results

### 3.1. Study Identification

The literature search strategy identified 862 references. After screening the title and abstract and removing the articles which fulfilled the exclusion criteria (Figure 2), 31 records underwent full-text review. After the full-text review, duplicates, non-randomized clinical trials and case reports were excluded. Finally, 13 records with a total sample size of 555 participants met the eligibility criteria.

The main characteristics of the trials included in the study are summarized in Table 1. The questions posed and the answers obtained in the qualitative analysis appear in Table 2. In the review the alloantibody data are organized into two groups (Table 3), eight studies measured alloantibodies with cPRA (calculated panel-reactive antibodies) [46,47,48,49,50,51,52,53,54] and five studies measured the presence of DSA [55,56,57,58,59].

Five studies carried out the measurement of alloantibodies after allo-MSC administration in patients with heart diseases, two studies in patients with ischemic cardiomyopathy (ICM) [46,54], two studies in patients with heart failure (HF) [55,56] and one study in patients with non-ischemic dilated cardiomyopathy (NDC) [47]. Two studies evaluated alloantibodies development in patients with diabetes, one in patients with type 2 diabetes [58] and the other in patients with diabetic neuropathy (DN) [53]. Two studies took this measurement in patients with osteoarthritis (OA) after allo-MSC treatment [48,49] and one study measured DSA in patients with rheumatoid arthritis (RA) [57]. Following joint disorders, one study carried out cPRA measurement after allo-MSC administration in patients with degenerative disc disease (DDD) [50,51] and another study carried it out in patients with frailty syndrome (FS) [52]. Finally, one study determined DSA in patients with Crohn’s disease (CD) after allo-MSC administration [59].

### 3.2. Bias Assessment

All the studies included a random sequence generation. Regarding the allocation concealment, two studies were considered to have some concerns as they did not specify the allocation methodology. There were some concerns for the blinding of participants and personnel in just one study and the blinding of outcome assessment was considered of high level of risk in two studies where masking of evaluators was single or none. Eleven studies conducted a blinding of outcome assessment to avoid the detection bias. We identified a high risk of attrition bias in three studies where the missing data proportion exceeded 20%. Ten studies managed properly the incomplete outcome data. Overall, the studies exhibited low risk of bias. The details of risk-of-bias evaluation for each included study are presented in Figure 3.

### 3.3. Outcomes

In the studies analyzed, a total of 555 patients have been entered into clinical trials with allo-MSC therapy. The main characteristics of the articles reviewed are summarized in Table 1. Table 2 presents the main questions posed as well as the answers obtained from the qualitative analysis.
jcm-10-02991-t001_Table 1Table 1Main Characteristics of included Randomized Clinical Trials (RTCs).First Author, Publish YearCountryIdentifierStudy DesignF/U (mo)Age Mean, (SD)Disease*n*CoTxCell Dose(×10^6^)InjectionsRoute of DeliveryMSC OriginHare, 2012 [54]United StatesNCT01087996RCT13Allo-MSCs62.8 (10.5)ICM30NoCohort AAuto-MSC (*n* = 5)Allo-MSC (*n* = 5)201TESIBMCohort BAuto-MSC(*n* = 5)Allo- MSC (*n* = 5)100Auto-MSCs63.7 (9.3)Cohort CAuto-MSC (*n* = 5)Allo- MSC (*n* = 5)200Skyler, 2015 [58]United StatesNCT01576328RCT24Co58.7 (7.3)Type 2 Diabetes6116Cohort AAllo-MSC(*n* = 15)0.3/kg1IVBMCohort A57.7 (8.2)Cohort BAllo-MSC(*n* = 15)1/kgCohort B55.3 (11.4)Cohort CAllo-MSC(*n* = 15)2/kgCohort C 57.2 (6.6)Perin, 2015 [55]AustraliaNCT00721045RCT12Co62.7 (11.2)Chronic HF6015Cohort AAllo-MSC(*n* = 15)251TESIBMCohort A60.1 (8.8)Cohort BAllo-MSC(*n* = 15)75Cohort B63.9 (11.5)Cohort CAllo-MSC(*n* = 15)150Cohort C62.7 (10.8)Vega, 2015 [48]SpainNCT01586312RCT12Co57.5 (9.5)OA3015Cohort AAllo-MSC (*n* = 15)401IABMCohort A56.6 (9.6)Panés, 2016 [59]Austria, Belgium, France, Germany, Israel, Italy, Netherlands, SpainNCT01541579 RCT24Co39.0 (13.1)CD11360Cohort AAllo-MSC (*n* = 63)1201ILATCohort A37.6 (13.1)Packham, 2016 [53]Mesoblast, AustraliaNCT01843387RCT15Co74.8 (7.9)DN3010Cohort AAllo-MSC (*n* = 10)1501IVBMCohort A70.5 (7.4)Cohort B64.8 (10.1)Cohort BAllo-MSC (*n* = 10)300Florea, 2017 [46]United StatesNCT02013674RCT12Cohort A 66.8 (12.2)ICM30NoCohort AAllo-MSC (*n* = 15)2010TESIBMCohort B 65.6 (9.4)Cohort BAllo-MSC (*n* = 15)100Tompkins, 2017 [52]United StatesNCT02065245RCT12Co75.3 (6.8)FS3010Cohort AAllo-MSC (*n* = 10)1001IVBMCohort A75.0 (7.4)Cohort BAllo-MSC (*n* = 10)200Cohort B76.3 (8.4)Álvaro-Gracia, 2017 [57]SpainNCT01663116RCT6Co58.43 (14.25)RA537Cohort AAllo-MSC (*n* = 20)1/kg3IVATCohort A54.15 (7.79)Cohort BAllo-MSC (*n* = 20)2/kgCohort B57.40 (11.01)Cohort CAllo-MSC (*n* = 6)4/kgCohort C50.33 (15.62)Hare, 2017 [47]United StatesNCT01392625RCT12Cohort A54.4 (11.5)NDC37NoCohort AAuto-MSC (*n* = 18)10010TESIBMCohort B 57.4 (11.0)Cohort BAllo-MSC (*n* = 19)Bartolucci, 2017 [56]ChileNCT01739777RCT12Co57.2 (11.6)HF3015Cohort AAllo-MSC (*n* = 15)1/kg1IVUCCohort A57.3 (10.1)Noriega, 2017 [51]SpainNCT01860417RCT1238 (2)DDD2412Cohort AAllo-MSC (*n* = 12)25/disc1IDBMWang, 2017 [49]AustraliaNCT01088191RCT24Co26.0 (3.6)OA176Cohort AAllo-MSC(*n* = 11)751IABMCohort A26.9 (10.3)Data summary of the reviewed clinical trials. Mo: month; SD: standard deviation; Co: control; Tx: treatment; MSC: mesenchymal stromal cell; Auto-MSC: autologous mesenchymal stromal cell; Allo-MSC: allogeneic mesenchymal stromal cell; RCT: randomized clinical trial; BM: bone marrow; AT: adipose tissue; UC: umbilical cord; TESI: transendocardial stromal cell injection; IV: intravenous injection; IL: intralesional injection; ID: intradiscal injection; IA: intraarticular injection; ICM: ischemic cardiomyopathy; HF: heart failure; CD: Crohn’s disease; DN: diabetic nephropathy; FS: frailty syndrome; RA: rheumatoid arthritis; NDC: nonischemic dilated cardiomyopathy; OA: osteoarthritis; DDD: degenerative disc disease; NE: not specifically.

-
What is the prevalence of alloantibodies development in patients treated with allo-MSCs?


Table 3 shows the development of alloantibodies in allo-MSC-treated patients. The presensitization status of some patients and its possible influence on treatment is also described. Of the 555 patients included in the analysis, 166 patients were treated as controls, 33 patients were administered auto-MSCs and specifically, 356 patients were administered allo-MSCs. 11.51% (41/356) patients developed alloantibodies. Specifically, 14.95% (32/214) developed DSA and 6.33% (9/142) developed cPRA. In 10 of the 13 studies included in the analysis, at least one patient developed alloantibodies.

-
Does the route of administration of allo-MSCs influence the development of alloantibodies?


The administration route for each study is shown in Table 1. The different routes of administration used were transendocardial stem cell injection (TESI), intradiscal (ID), intra-articular (IA), intralesional injection (IL) and intravenous (IV) injections, depending on the disease treated. The most common route of administration was IV, 38% (5/13). The second most repeated route of administration was TESI with four studies. Two studies were carried out using route IA. Both IL and ID were only used in one study each.

TESI injection in all studies led to the development of alloantibodies. The values were variable, ranging from one single patient [46,47], two patients [54], and five patients [55]. IV injection was also variable, in two trials it did not produce DSA development [56,58]. In the rest of the studies in which IV injection was used, there was a DSA response. The results found 5% (1/20) [53], 15% (3/20) [52] and 19% (9/48) [57] of the patients with DSA development in the different studies. Both studies that carried out the IA injection produced cPRA that decreased over time [48,49,50]. IL injection produced the highest production of DSA. Specifically, 34% of patients who received a dose of allo-MSCs generated DSA [59]. In the only study in which allo-MSC was administered via ID, no alloantibodies development was observed in any treated patient [50,51].

-
Does the tissue of origin of the MSCs influence the development of alloantibodies?


The tissue origin of the allo-MSCs is indicated in Table 1. Bone marrow (BM), umbilical cord (UC) and adipose tissue (AT) were used as tissue origin for MSCs. When allo-MSCs were obtained from UC, there was no development of DSA in any treated patient [56]. When cells were obtained from AT, the greatest DSA response occurred (specifically 34% of CD patients treated with allo-MSC [59] and 19% in RA patients treated with allo-MSC [57]). Finally, when the origin was BM, the results were variable. In eight of the ten studies in which allo-MSC derived from BM were used, alloantibodies development occurred [47,48,49,50,52,53,54,55], and only two studies of these ten did not reported any patient with DSA [50,51,58].

-
Is the total dose of allo-MSCs associated with increased alloantibodies response?


The cell dose is described in Table 1. The lowest dose of allo-MSCs administered was 0.3 M/kg of body weight in patients with type 2 diabetes. This dose did not cause the development of DSA in any treated patient [58]. In another study in which the cell dose was low, specifically 1 M, no patient developed DSA [56]. Furthermore, Noriega et al. administered 25 M/disc and no alloantibodies development was found [50,51].

On the other hand, the highest dose was 300 M to treat DN. In this case, two patients developed alloantibodies [53]. Panes et al. administered 120 M leading to a frequency of DSA development of 34% [59]. Hare et al. with a cell dose of 100 M found a 30% development of alloantibodies [47]. Álvaro-Gracia et al. administered doses of 1 M, 2 M and 4 M, and 19% of the patients developed DSA [57]. They did not indicate differences between the administered doses and their relationship with the development of DSA.

Perin et al. compared different doses to treat chronic HF. The response of DSA persisted longer in patients treated with the highest dose. Specifically, the antibody response was transient in three patients (two patients in the 25 M-treated group and one patient in the 150 M-treated group). However, the response of DSA persisted for ≤12 months in two patients in the 150 M-treated group [55].

Tompkins et al. also compared different doses, in this case to treat FS. The highest production of alloantibodies occurred in patients treated with the highest dose. Specifically, three patients had a mild/moderate increase in cPRA (one mild in the 100 M-treated group and two moderates in the 200 M-treated group) [52].

-
Are repeated doses of treatment associated with higher frequency of alloantibodies or impact of treatment?


The number of allo-MSC injections is described in Table 1. In most studies (10 of 13) a single infusion of allo-MSCs was performed. Álvaro-Gracia et al. performed three allo-MSC infusions to treat RA [57]. It is one of the studies included in this systematic review where the highest production of DSA was observed (19% of the allo-MSC treated patients). Hare et al. and Florea et al., performed 10 allo-MSC infusions to treat nonischemic DC and ICD, respectively [46,54]. In both trials, only one patient developed alloantibodies. When the infusion was unique, the data is highly variable.

-
Does the presensitization status of patients influence the development of alloantibodies?


The patient’s state of presensitization is described in Table 3. Of the 13 studies analyzed, four indicated the presence of presensitized patients. Usually presensitized patients were prone to a sustained humoral response [59]. In one of these studies the presence of presensitized patients was not directly indicated, however, there were two patients in whom the DSA response persisted ≤12 months, which may be due to this state [55]. Hare et al. detected that more than 30% of patients tested showed sensitization to HLA antigens at baseline. A majority of the sensitized patients (87.5%) demonstrated sensitization at all time points with minimal variation of antibody levels [54]. Alvaro-Garcia et al. Showed the largest number of presensitized patients. Specifically, 43% of treated patients presented baseline anti-HLA-I antibodies. Presensitized patients showed higher frequency of DSA (30% vs. 11%) [57].

-
How relevant is alloantibodies development to the safety of the treatment?


Adverse events (AEs) and severe adverse events (SAEs) collected in the clinical trials analyzed are summarized in Table 3.

In none of the analyzed studies was there a correlation established between the safety profile and the development of alloantibodies. The most frequently reported AEs in patients included in the analysis were infections, which are indicated as problems related to the injection process and not to the drug [52,57,58]. The production of alloantibodies does not influence the AEs or SAEs appearance.

Only two studies [48,50,51] analyzed the number of mismatches between donor and recipient. However, the number of mismatches was not correlated with the generation of alloantibodies, nor did it have clinical consequences.
jcm-10-02991-t002_Table 2Table 2Main research questions posed, and results obtained in the qualitative analysis.Research QuestionAnswerWhat is the prevalence of alloantibodies’ development in patients treated with allo-MSCs?10.67% of patients developed alloantibodiesDoes the route of administration of allo-MSCs influence the development of alloantibodies?IL injection produced the highest production of alloantibodies ID injection did not produce any alloantibodiesDoes the tissue of origin of the MSCs influence the development of alloantibodies?MSC-AT produced the highest production of alloantibodies MSC-UC did not produce any alloantibodiesIs the total dose of allo-MSCs associated with increased alloantibodies response?Higher doses of allo-MSCs are generally associated with increased development of alloantibodiesAre repeated doses of treatment associated with higher frequency of alloantibodies or impact of treatment?No correlation was established between the development of alloantibodies and the number of doses of allo-MSCsDoes the presensitization status of patients influence the development of alloantibodies?Presensitized patients generally showed the highest frequency of alloantibodiesHow relevant is alloantibodies development to the safety of the treatment?No correlation was established between the development of alloantibodies and the safety of treatment with allo-MSCs
jcm-10-02991-t003_Table 3Table 3Summary of the efficacy and adverse events associated with the generation of alloantibodies by Mesenchymal Stromal Cell therapy.First Author, Publish YearNº PatientsDiseaseAdverse Events (AEs) and Severe Adverse Events (SAEs)EfficacyAlloantibodiesCorrelation Antibodies + Adverse EventsPrincipal EndpointsResultsMeasurementResultsHare, 2012 [54]30ICM30 days, one patient in each group was hospitalized for heart failure(TE-SAE rate 6.7%)The 1-year incidence of SAEs was 33.3% in the allo-MSC and 53.3% in the auto-MSC (*p* = 0.46).At 1 year, there were no ventricular arrhythmia SAEs observed among allo-MSC recipients compared with four patients (26.7%) in the auto-MSC group (*p* = 0.10).6 MWT,exercise peak VO2, MLHFQ,LV volumes,EF,EEDAuto-MSC but not allo-MSC therapy was associated with an improvement in the 6 MWT and the MLHFQ score, but neither improved exerciseVO2 max.Allo-MSCs and auto-MSCs reduced mean EED by −33.21% and sphericity index but did not increase EFAllo-MSCs reduced LV end-diastolic volumes.Low-dose concentration MSCs produced greatest reductions in LV volumes and increased EF.cPRAMore than 30% of patients tested showed sensitization to HLA antigens at baseline. A majority of the sensitized patients (87.5%) demonstrated sensitization at all time points with minimal variation of antibody levels.Two patients in the allo-MSC group showed sensitization only at the 6-month time point. Of these, one patient developed low-level HLA class I antibodies to HLA antigen specificities not expressed by the donor MSC. The other sensitized patient showed low-level donor-specific HLA class I antibodies.Not foundSkyler, 2015 [58]61Type 2 Diabetes27 AEsWeek 12: No SAEs, serious hypoglycemia AEs, or discontinuations due to AEs were found.HbA1 cWeek 12: The HbA1c target of 7% was achieved in 8 of 45 subjects treated with 2M-MSCs versus 0 of 15 subjects treated with placeboDSANo subjects developed antibodies specific to the donor HLANot foundPerin, 2015 [55]60Chronic HFThe incidence of AEs was similar across all groups.Three AEs associated to a procedure in the 150 M.Two episodes of sustained ventricular tachycardia in treated patients (one patient in the 25 M and one in the 75 M).MACE was seen in 15 patients: 10 of 45 (22%) MSC-treated and 5 of 15 (33%) control patients.Two cardiac deaths in the MSC groups (4.4%), both in the M and three in the control group (20%).LVESVCompared with the control group, the 150 M-MSC group showed improvement in LVESV with a statistically significant decrease at 6 months (*p* = 0.015) and a non significant decrease at 12 months.There were no consistent differences between the treated and control groups in myocardial perfusionDSA11% of the 45 M treated patients developed DSAThis response was transient in three patients (two 25 M-treated patients and one 150 M-treated patient) but persisted for ≤12 months in two patients in the 150 M group.Not foundVega, 2015 [48]30OA48 AEs (25 in the control group and 23 in treated patients).No SAEs occurred during treatmentVAS,Lequesneindex,WOMAC indexPCIValues of all evaluation scalesimproved with the cell treatment.Improvement of patients was medium to large (effect size, 0.58 to 1.12for the different algofunctional indices), whereas improvement was small (0.19 to 0.48) after control treatment.The PCI decreased in bothgroups, but the decrease was not statistically significant in the control, whereas it reached significance (*p* < 0.05) in the experimental group at the 12-month follow up.Pain was significantly reduced by 6 and 12 months after MSC treatment.In the treated patients, the slope of the line (efficiency of treatment) was 0.69, whereas in the control series, the slope was only 0.28.77% of the patients were satisfied or very satisfied with the treatment, whereas in the control group, this percentage fell to 38%.cPRASpecific anti-HLA antibodies targeted to alleles present in the donor were found in only 2 of the 13 patients assessed during the trial. In these patients, the reactivity decreased with time.Not foundPanés, 2016 [59]212CD17% of patients in the MSC-treated group versus 29% in the placebo groupexperienced TE-AEs, the most common of which were anal abscess and proctalgia.TE-SAEs reported were 5% in the MSC group vs. 7% in the placebo group.Clinical remissionResponseTime to relapsePDAI50% clinical remission in MSC-treated patients (53 of 107) versus 34% remission in placebo-treated patients (36 of 105).Time to remission was significantly shorter in MSC-treated patients (6–7 weeks) than in the placebo group (14 weeks).The improvement in PDAI with MSCs was significantly greater than with placebo at week 6 (change from baseline treatment difference −1.0, 95% CI −1.7 to −0.3), week 12 (−1.2, −2.0 to −0.4) and week 18 (−1.2, −2.0 to −0.3), but not at week 24 (−0.8, −1.8 to 0.2).DSA16% MSC-treated patients and 15% placebo-treated patients were sensitized34% of MSC-treated patients and none of the placebo-treated patients generated DSA.Presensitized patients were prone to a sustained humoral response longer.Not foundPackham, 2016 [53]3012 weeksSeven (70%) of TE-AEs were reported in the placebo group,Eight (80.0%) in the 150 M groupNine (90.0%) in the 300 M group.The most commonly were edema peripheral, lower respiratory tract infection, urinary tract, infection, cataract, and anemia.No acute allergic or immunologic adverse events were reported.TE-SAEs wereTwo (20.0%) in the placebo group,four (40.0%) in the 150 M,one (10.0%) in the 300 M.eGFRmGFRThe adjusted least squares mean differences from placebo in changes from baseline in the treated-groups were 4.4 ± 2.16 and 1.6 ± 2.15 mL/min/1.73 m^2^ for eGFR and 4.1 ± 2.75 and 3.9 ± 2.75 for mGFR for the 150 M and 300 M groups, respectively.cPRATwo treated-patient developed DSA.One placebo-treated patient was sensitized.Not foundFlorea, 2017 [46]3012 monthsThe incidence of AE was10 (66.7%) in the 20 M group and 13 (86.7%) in the 100 M group.The incidence of SAE was seven (46.7%) in the 20 M group and five (33.3%) in the 100 M group.MACE rate was 20.0% in the 20 M group and 13.3% in the 100 M group.Worsening heart failure rehospitalization was20.0% in the 20 M group and 7.1% in the 100 M group.One case of death in the 100 M groupScar sizeEFScar size was reduced to a similar degree in both groups: 20 M by −6.4 g (interquartile range, −13.5 to −3.4 g; *p* = 0.001) and 100 M by −6.1 g (interquartile range, −8.1 to −4.6 g; *p* = 0.0002), the EF improved only with 100 M by 3.7 U (interquartile range, 1.1 to 6.1; *p* = 0.04cPRAOnly one patient in the 20 M treated group had no- to low-cPRA response.Not foundTompkins, 2017 [52]30AF10 AEs (four in MSC-treated patients and six AEs in placebo-treated patients).One SAE (death) in MSC-treated group was reported (non-treatment related)Physical performance: 6 MWT,SPPB score, FEV1.Immune biomarkers6 MWT increased in the 100 M-groupfrom baseline to 6 months (345.9 ± 103.4 to 410.7 ± 155.4 m, *p* = 0.011)SPPB total score was significantlyimproved in the 100 M-group from baseline to 6 months (median10.5, IQR 9.0, 12.0 to 12.0, IQR 11.0, 12.0; *p* = 0.031)FEV1 improved in the 100 M-group from baseline to 6 months (2.5 ± 0.66 to 2.6 ± 0.77 L/min, *p* = 0.025)Serum TNF-α levels decreased in the 100 M-group (*p* = 0.03). B cell intracellular TNF-α improved in both the 100 M- (*p* < 0.0001) and 200 M-groups (*p* = 0.002) as well as between groups compared to placebo (*p* = 0.003 and *p* = 0.039, respectively)cPRAThree patients had a mild/moderate increase in donor specific antibodies(one mild in the 100 M- and two moderates in the 200 M-group).There were no clinically significant immune reactions reported.Not foundÁlvaro-Gracia, 2017 [57]5324 weeks141 AEs (most frequent were fever (17%) and infections (15%)).85% of 1 M-MSC group,75% of 2 M-MSC group,100% of 4 M-MSC group and57% of the placebo group experienced at least one AE.Five SAEs in 1 M-MSC group: lacunar infarction, diarrhea, tendon rupture, rheumatoid nodule and arthritisTwo SAEs in 2 M-MSC group: sciatica and rheumatoid arthritisOne SAE in placebo: astheniaProportionof ACR20, ACR50, ACR70.ACR20 responses for cohorts A, B, C and placebo were 45%, 20%, 33% and 29%, respectively, at month 1; 25%, 30%, 17% and 14%, respectively, at month 2; and 25%, 15%, 17% and 0%, respectively, at month 3.ACR50 responses tended to be generally greater in patients treated whereasACR70 responses were very low.DSA19% of patients generated DSA without apparent clinical consequences.43% of treated patients presented baseline anti-HLA-I antibodies (presensitized)Presensitized patients showedhigher frequency of DSA (30% vs. 11%).Not foundHare, 2017 [47]37NDCThe 12-month post-TESI AE incidence was 66.7% in allo-MSC and 87.5% in auto-MSC patients.The 12-month post-TESI SAE incidence was 28.2% in allo-MSC and 63.5% in auto-MSC patients.LV structure andfunction,QOL,6 MWT,FEV1,endothelial functionEPC-CFUFMBEF increased significantly in the allo-MSCs group by 8.0 percentage points (*p* = 0.004), but not in the auto-MSCscohort (*p* = 0.116) at 12 months.Functional capacity and QOL showed greater improvement with allo- compared with auto-MSCs use.The 6 MWT distance significantly increased in patients receiving allo-MSCs by 37.0 m (*p* = 0.04) at 12 months compared with baseline, but did not significantly change in the auto-MSCs group (*p* = 0.71).FEV1 improved in allo-MSCs patients by 3.7% (*p* = 0.2423) compared with a decrease of 3.8% (*p* = 0.16) among the auto-MSCs group at 12 months.EPC-CFU significantly increased with allo-MSCs (*p* = 0.0107) as did the percentage of FMD at 3 months (*p* = 0.09).cPRAOne allo-MSC patient developed an elevated (>80%) donor-specific cPRA levelNot foundBartolucci, 2017 [56]30HFThere were no acute AEs associated with the infusionOf allo-MSCs or placebo28 clinically relevant events were reported and the most frequent were non sustained ventricular tachycardia (seven in the placebo group and seven in the treated group)LVEF,NYHA functional classification,ventilatory efficiency (VE/VCO2 slope)Only the MSC- treated group exhibited significant improvements in LVEF at 3, 6 and 12 months of follow-up assessed both through transthoracic echocardiography (*p* = 0.0167 versus baseline) and cardiac MRI (*p* = 0.025 versus baseline).Echocardiographic LVEF change from baseline to month 12 differed significantly between groups (+7.07 ± 6.22% versus +1.85 ± 5.60%; *p* = 0.028).MSC- treated patients displayed improvements of NHYA functionalclass (*p* = 0.0167 versus baseline) MSC-treated exhibited an improvement in VE/VCO2 at 12 months (−1.89 ± 3.19; *p* = 0.023 versus baseline)DSANone of the patients tested presented alloantibodies to the MSCsNot foundNoriega, 2017 [51]24DDDOccasional mild pain reactionsPain and disabilityindexesVASODISF-12 life quality questionnairePfirrmann gradingMSC-treated patients displayed a quick and significant improvement in algofunctional indices vs. the controls. This improvement seemed restricted to a group of responders that included 40% of the cohort.Degeneration improved in the MSC treated patients and worsened in the controls.cPRASpecific antibodies were not detected in any of the nine patients tested.Not foundWang, 2017 [49]17OA133 AEsMost frequent were musculoskeletal and connective tissue disorders (11 in MSC group and six in control group)Two SAEs were reported in MSC group (fracture of the humerus andinfective bursitis). These were not consideredto be treatment related.KOOSSF-36v2 scoresJoint space widthtibial cartilage volumebone areaCompared with the control group, MSC-treated patients showed greater improvements in KOOS pain, symptom, activities of daily living and SF-36 bodily pain scores (*p* < 0.05).26 weeks: The MSC group had reduced medial and lateral tibiofemoral joint space narrowing (*p* < 0.05), less tibial bone expansion (0.5% vs. 4.0%, *p* = 0.02), and a trend towards reduced tibial cartilage volume loss (0.7% vs. −4.0%, *p* = 0.10) than the controls.cPRAIncreases in anti-HLA (class I) PRA >10% were observed at week 4 in the cell group that decreased to baseline levels by week 104Not foundAE: adverse event; SAE: severe adverse event; 6MWT: 6-min walk test; MLHFQ: Minnesota Living with Heart Failure Questionnaire; NYHA: New York Heart Association class; LV: left-ventricular; ef: ejection fraction; EED: early enhancement defect; TE-SAE: treatment emergent-severe adverse event; LVESV: left ventricular end systolic volume; MACE: major adverse cardiac events; PCI: poor cartilage index; KOOS: Knee Injury and Osteoarthritis Outcome Score; PRA: panel reactive antibodies; VAS: visual analogue scale; ODI: oswestry disability index; SF-12: short form-12; LVEF: left ventricular ejection fraction; EF: ejection fraction; QOL: quality of life; EPC-CFU: endothelial progenitor cell colony forming unit; FMD: flow-mediated vasodilation; FEV1: forced expiratory volume in 1 s; ACR: American College of Rheumatology; SPPB: short physical performance battery; PDAI: perianal disease activity index; eGFR: estimated glomerular filtration rate; mGFR: measured glomerular filtration rate; cPRA: calculated panel reactive antibodies; ‘’+’’: symbol indicated to establish correlation between two variables.

## 4. Discussion

In this systematic review, the production of alloantibodies after treatment with allo-MSCs was not correlated with safety or tolerability and did not have clinical relevance. Although a better understanding of MSC rejection is still needed, it is clear that MSCs are not immunoprivileged, but rather evade the immune system [15]. However, this has not influenced their application, and their use in clinical trials continues to increase. In fact, evidence from currently completed and ongoing clinical trials demonstrate that allo-MSC treatment is safe and does not appear to cause significant patient side effects [3]. The administration route of MSCs, the tissue origin, as well as the total cellular dose administered are parameters that can influence the development of alloantibodies. It is critical to note that although infused MSCs may not express class II MHC, this is likely to be activated in vivo at sites of inflammation [60]. Immunogenicity should be recognized as a characteristic of MSCs and its impact on MSC therapy should be examined. Therefore allo-MSC therapy faces significant challenges.

-
What is the prevalence of alloantibodies development in patients treated with allo-MSCs?


One out of eleven patients developed alloantibodies after allo-MSCs administration. In all the studies, specificity of DSA was against HLA class I antigens. Local treatments seem to have a lower biodistribution and as they do not have systemic biodistribution [61]. However, in our review, the IL injection was the one that produced the highest generation of alloantibodies [59]. The presence of DSA could cause the loss of efficacy because of immune clearance of the product. However, Perin et al. [55] found the only sustained titers of DSA in the higher dose group (150 million cells), which had the best sustained efficacy response. On the other hand, Hare et al. [54] and Tompkins et al. [52] found that the lowest doses had better endpoints but this was not correlated with the presence of alloantibodies. In general, the highest doses of MSCs appear to be associated with greater production of the DSA response.

-
Does the presensitization status of patients influence the development of alloantibodies?


Regarding the presence of presensitized patients, in the studies in which presensitization is analyzed, the percentage was 43% [57], 30% [54] and 16% [59]. Possible explanations for a non-exposed patient developing alloantibodies include some sensitizing event such as a blood transfusion, vaccination, infection or exposure to some other unidentified antigen. Infection could upregulate the immune system resulting in expression of these antibodies [53]. The presensitization status of patients seems to produce a more rapid and sustained response over time to DSA development. However, the presence of DSA at baseline did not have clinical relevancy or affected safety and tolerability. Although the number of MSCs infusions may produce presensitization, this is not related with the DSA development.

Presensitized patients showed the highest frequency of DSA, but this did not have clinical relevance in the studies analyzed. Because of the lack of studies, it is difficult to draw a firm conclusion. If the development of DSA proves to be of clinical importance, MSCs from representative subsets of the population could be collected, typed, and expanded. The creation of an HLA typed allogenic MSC bank could be created due to MSCs expansion. This should improve MSC persistence and potentially therapeutic outcomes. It should also allow serial injection of MSCs without rejection or co-administration of immunosuppressive drugs [15].

-
Does allo-MSC treatment influence the safety of the treatment?


Regarding the analyzed studies, the safety profile and the de-novo DSA development were not correlated. Although MSC therapy shows a favorable safety profile, long-term safety data are limited. It would be interesting to carry out a longer follow-up after this therapy in the safety profile, as well as to evaluate the possible long-term effects that the development of alloantibodies may have. In fact, Barnhoorn et al. [21] detected an Epstein-Barr virus [EBV] associated with a B cell lymphoproliferative lesion in the rectum of a patient 4 years after local administration of MSCs for his perianal fistulas. It is possible that an additional immunosuppressive effect was produced in patients also receiving immunosuppressive medication. MSC therapy can induce additional local immunosuppression, which can subsequently drive proliferation of tissue-resident EBV-infected cells. Furthermore, Ringdén et al. [62] and Zhao et al. [63] describe two cases of EBV-associated lymphoproliferative disease after systemic treatment with MSCs in patients with GHVD. However, more long-term reports on MSC therapy in clinical trials and daily practice are needed to assess the full safety profile of MSC therapy.

-
What information do non-randomized studies give us?


In our search, we found two case reports and six non-randomized clinical trials in which DSA was also measured after allo-MSC therapy. These were not included in this systematic review because they were non-randomized studies. Despite the fact that risk of alloimmunization by MSCs seems to be low in immunocompromised patients, numerous cases have been found that point out the opposite. For example, in the case report of Kaipe et al., after two infusions of MSCs, the severe junctional epidermolysis bullosa patient had developed multispecific anti-HLA class-I DSA [64]. Sun et al. [65] detected new class I anti-HLA DSA in 1/3 participants who received one dose of MSCs, 1/3 participants who received two doses, and 3/6 participants who received three doses. Participants who developed broad-spectrum DSA were treated with the same batch of MSC, indicating that some HLA haplotypes may be more antigenic than others. Furthermore, all participants who were at least haploidentical to their MSC donor developed new-onset DSA. This may be due to the use of low-resolution technologies to typify HLA. These antibodies have not been associated with any clinical manifestations to date. Clé et al. [60] found that patients who remained refractory to aplastic anemia had at least one HLA class I DSA. Two refractory patients had an HLA class II-driven antibody from a donor, supporting the idea that HLA class II from MSCs can be expressed at sites of inflammation.

### 4.1. Limitations

Only two studies [48,50,51] analyzed the number of mismatches between donor and recipient and just four studies evaluated the presensitization of the patients. Furthermore, the previous immunosuppression status of the patients was not considered. These parameters are crucial, and they may be related to the alloantibodies’ development after allo-MSC treatment. Another factor which may affect the immunogenicity of MSCs is their heterogeneity. Donor, source and sample collection as well as isolation and culture techniques can influence MSCs phenotype and immunomodulatory properties [66,67]. For instance, regarding culture conditions, MSCs cultured in human platelet lysate showed impaired inhibitory capacity on T-cell proliferation to alloantigen and NK-cell proliferation and cytotoxicity [67,68]. Furthermore, cryopreserved MSCs for less than 6 months have been reported significantly more suppressive than both fresh MSC and cells stored for longer time periods [69]. Cell quality may also affect MSCs function. El Sayed et al. reported that late-passaged BM-MSCs released less TGF-β than early-passaged BM-MSCs [70]. Therefore, the functional heterogeneity of MSCs must be taken into account in the clinical trials to optimize MSC-based therapy. A consensus about well-defined culture methods may help standardization of MSCs production.

### 4.2. Recommendations for Futures Studies

In only 13 studies in which allo-MSCs were used as treatment, was the production of alloantibodies measured. Of these, just four studies evaluate the presence of DSA at baseline and two studies analyzed the number of HLA mismatches between donor and recipient. In future studies it may be interesting to analyze these parameters in order to extract firm conclusions about the immunogenicity of allo-MSC treatment. Importantly, anti-donor antibody measurements in clinical trials, which to date have been rare, should become routine, as they will allow to understand the relationship between the degree of HLA mismatch, MSC rejection, and the efficacy of treatment under specific conditions [15]. In addition to the measurement of DSA, functional assays should be performed to determine the type of immune responses and to assess the possible implications for clinical therapy. In addition, testing at multiple time points after administration is recommended to measure the kinetics of the immune response. The mean assays used for measuring cell-mediated and humoral immune response against allo-MSCs include ex vivo mixed lymphocyte reactions (MLR), enzyme-linked immunospot (ELISPOT), antibody-dependent complement-dependent cytotoxicity (CDC) and in vivo imaging [24]. Only Perin et al. analyzed the activation of T-cells through MLR showing that MSCs did not elicit proliferative responses from patients’ cells [55].

## 5. Conclusions

In conclusion, parameters such as the administration route, the tissue origin or the size of the MSC dose can influence the subsequent development of alloantibodies. The IL administration route produced the greatest generation of alloantibodies. Allo-MSCs from UC did not produce any patient with alloantibodies, and high doses are generally associated with a high concentration of alloantibodies.

Furthermore, no correlation between alloantibodies and the safety of allo-MSC therapy has been found. However, only 13 studies in which allo-MSCs were used as treatment measured the production of alloantibodies. This may be insufficient to extract firm conclusions about the immunogenicity of allo-MSC treatment and more studies are needed.

## Figures and Tables

**Figure 1 jcm-10-02991-f001:**
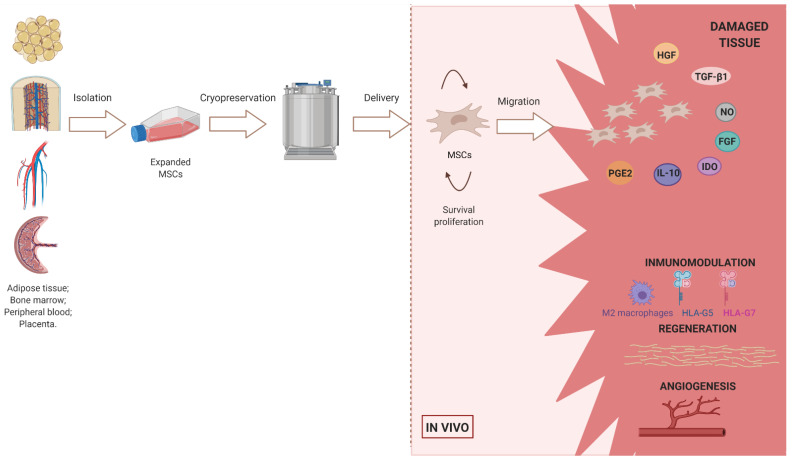
This scheme summarizes the entire MSC manufacturing process for allogeneic cellular therapy and its subsequent function in the recipient tissue. The secretion of cytokines and other factors such as NO, IDO, HGF, FGF, IL-10, PGE2 and TGF-β1 contribute to the tissue repair processes that can be summarized in three points: (1) Immunomodulation; (2) Regeneration; and (3) Angiogenesis. Created with BioRender.com.

**Figure 2 jcm-10-02991-f002:**
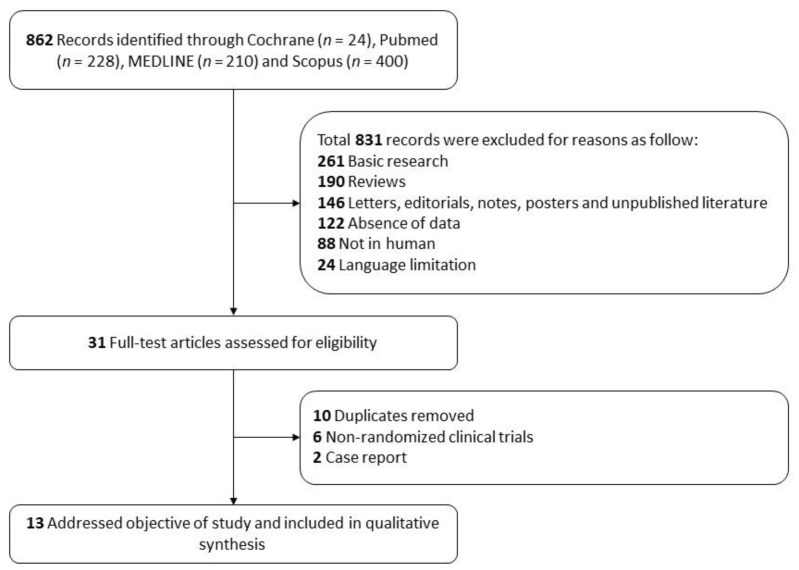
Flow diagram of the study selection process for inclusion in the systematic review according to the Preferred Reporting Items for Systematic Reviews and Meta-Analysis (PRISMA).

**Figure 3 jcm-10-02991-f003:**
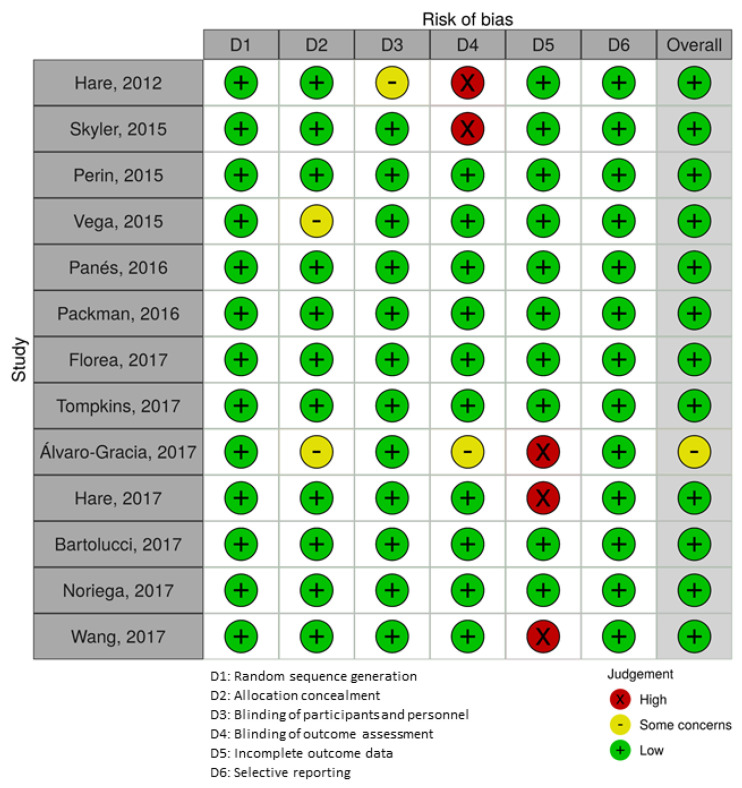
Quality assessment of the included studies. The risk-of-bias summary is based on the review authors’ judgements about each risk-of-bias item for each individual study: The green circle indicates low risk, and the red circle indicates high risk. Some concerns risk is represented by the yellow circle.

## Data Availability

Data is contained within the review.

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
