# Peer review of "Alloreactive Immune Response Associated to Human Mesenchymal Stromal Cells Treatment: A Systematic Review"

_jcm, 2021, doi:10.3390/jcm10132991_

Round 1

Reviewer 1 Report

In this manuscript the authors perform a meta-analysis of 13 clinical trials that investigate on the alloreactive immune response against mesenchymal stromal cells (MSCs). The topic is original and of significant interest to the scientific community. The manuscript is well-written and, except for the introduction, is well-organized. But there are several issues that should be solved to increase the quality of the text:

1) One important variable that have not been accounted  for by the authors is the difference in the culturing methods of MSCs. The authors admit in the intro that differences in the culturing methods (e.g. use of activation of MSCs by pro-inflammatory stimuli) may affect the ability of MSCs to evade the immune system. Also, it is well-known in the field that the ability of MSCs to orchestrate the immune system sharply declines with time if the cells are cultured in vitro in standard conditions. Therefore, both the number of doubling before transplantation into patients and small changes in the handling technics might affect the immunogenecity of MSCs. The authors should add a paragraph on the problem of variability of the culturing time and methods in the Limitations chapter.

2) Acute Respiratory Distress Syndrome is a highly inflammatory conditions and of particular interest with regard to the current Covid-19 pandemics. The authors should add information and references on treatment of ARDS using MSCs to the Introduction.

3) The Introduction lacks structure and should be reorganized. For instance, the authors first describe the immunogenecity of allogenic MSCs then clinical trials with MSCs then general approaches for use of MSCs then the immunogenecity of MSCs again.

4) The sentence in lines 45-46 is not accurate, because technically MSCs can be derived from the majority of organs and tissues of the body. The sentence should be rewritten.

Author Response

We would like to thank you for taking the time and effort necessary to review our manuscript.

We sincerely appreciate all valuable comments and suggestions, which helped us to improve the quality of our review.

Corrections appear in blue font in the text.

Point 1.  One important variable that have not been accounted for by the authors is the difference in the culturing methods of MSCs. The authors admit in the intro that differences in the culturing methods (e.g. use of activation of MSCs by pro-inflammatory stimuli) may affect the ability of MSCs to evade the immune system. Also, it is well-known in the field that the ability of MSCs to orchestrate the immune system sharply declines with time if the cells are cultured in vitro in standard conditions. Therefore, both the number of doubling before transplantation into patients and small changes in the handling technics might affect the immunogenecity of MSCs. The authors should add a paragraph on the problem of variability of the culturing time and methods in the Limitations chapter.

Information about the variability of the culture time and methods has been added to the Limitation chapter (line 500):

“Another factor which may affect the immunogenicity of MSCs is their heterogeneity. Donor, source and sample collection as well as isolation and culture techniques can in-fluence MSCs phenotype and immunomodulatory properties [68,69]. For instance, re-garding culture conditions, MSCs cultured in human platelet lysate showed impaired inhibitory capacity on T-cell proliferation to alloantigen and NK-cell proliferation and cytotoxicity [69,70]. Furthermore, cryopreserved MSCs for less than 6 months have been reported significantly more suppressive than both fresh MSC and cells stored for longer time periods [71]. Cell quality may also affect MSCs function. El Sayed et al reported that late-passaged BM-MSCs released less TGF-β than early-passaged BM-MSCs [72]. Therefore, these functional heterogeneity of MSCs must be taken into account in the clinical trials to optimize MSC-based therapy. A consensus about well-defined culture methods may help standardization of MSCs production”

Point 2. Acute Respiratory Distress Syndrome is a highly inflammatory conditions and of particular interest with regard to the current Covid-19 pandemics. The authors should add information and references on treatment of ARDS using MSCs to the Introduction.

Information about MSCs therapy to treat ARDS has been added to the Introduction (line 130):

“Recently, the role of MSCs therapy in acute respiratory distress syndrome (ARDS) ad-dressed to severe cases of COVID-19 infection has been studied. In these patients, MSCs have demonstrated safety and possible efficacy [40] because of their anti-inflammatory, antiapoptotic, anti-microbial, and pro-angiogenic effects [41,42]. MSCs also promote bacterial and alveolar fluid clearance and disrupt the pulmonary endothelial and epithelial cell damage as they are capable of transferring mitochondria to injured epithelial cell [40]. Furthermore, angiopoietin-1 and keratinocyte growth factor secretion con-tribute to the restoration of alveolar–capillary barriers [43]. Regarding clinical trials, improved radiographic findings, pulmonary function and inflammatory biomarker levels have been found [41]”.

Point 3. The Introduction lacks structure and should be reorganized. For instance, the authors first describe the immunogenecity of allogenic MSCs then clinical trials with MSCs then general approaches for use of MSCs then the immunogenecity of MSCs again.

The Introduction has been reorganized. Firstly, definition of MSCs and their origin are described. Secondly, low immunogenicity and the immunomodulatory properties of MSCs are indicated. Thirdly, preclinical studies providing evidence about immunogenicity of MSCs are described. Finally, the need to analyze the immunogenicity of allogeneic MSCs in clinical studies is highlighted as well as the aim of the systematic review.

Point 4. The sentence in lines 45-46 is not accurate, because technically MSCs can be derived from the majority of organs and tissues of the body. The sentence should be rewritten.

The sentence in lines 45-46 has been rewritten:

“Adult MSCs can be obtained from different tissues although, the most widely used are bone marrow, peripheral blood or adipose tissue of patients [4]”.

Reviewer 2 Report

In this systematic review the authors collect and describe the available data on antibody generation after allogeneic MSC therapy in randomized clinical trial and evaluate its potential impact on safety and efficacy.

Comments

- I would suggest to focus the manuscript only on the effect of antibody generation, if any, as a safety and tolerability issue of the MSC therapy. Indeed, almost all the clinical trials included in the analysis are of phase1/2, therefore not powered to assess the efficacy.

On the other hand, data on the impact of alloantibodies against MSC on the efficacy of the cell therapy in preclinical models (described in lines 76-109), should be better detailed, including also studies with MSC not stimulated by pro-inflammatory cytokines.

-  A close attention should be payed when describing data on alloantibodies. Some studies reported the level of panel reactive antibodies (cPRA) which are not necessarily donor-specific (for example, refs 39 and 44). Similarly, presence of anti-HLA antibodies (not always the HLA expressed by allogeneic MSC) before the injection of allogeneic MSC could indicate pre-sensitization toward allogeneic HLA. This situation should be discussed separately from de-novo DSA development against MSC, since pre-formed antibodies can destroy the infused cells, while de-novo generation of MSC donor-specific antibodies can eliminate these cells in case of repeated injections. In addition, de-novo generation of MSC donor-specific antibodies can expose the patient to some risks, such as the low accessibility to a future organ transplantation.

-  Ref 43 and 46 did not report any data on DSA development, please address this inconsistency.

-  In ref 50 only a subgroup of patients given allo-MSC had anti-HLA antibodies measured. Please re-calculate and provide the exact number of patients analyzed overall.

-  Line 128-130. The sentence appears to be incorrect. The generation of DSA is likely the results of indirect recognition of MSC HLA presented by patient APC to CD4+ T cells.

Author Response

We would like to thank you for taking the time and effort necessary to review our manuscript.

We sincerely appreciate all valuable comments and suggestions, which helped us to improve the quality of our review.

Corrections appear in red font in the text.

Point 1. I would suggest to focus the manuscript only on the effect of antibody generation, if any, as a safety and tolerability issue of the MSC therapy. Indeed, almost all the clinical trials included in the analysis are of phase1/2, therefore not powered to assess the efficacy. On the other hand, data on the impact of alloantibodies against MSC on the efficacy of the cell therapy in preclinical models (described in lines 76-109), should be better detailed, including also studies with MSC not stimulated by pro-inflammatory cytokines.

We have focused the manuscript on the effect of antibody generation on the safety and tolerability of allo-MSC treatment. Therefore, the correlation of alloantibody development with the efficacy of MSC treatment has been removed from the entire manuscript.

On the other hand, data on the impact of alloantibody development on the efficacy of MSC treatment in preclinical studies has been expanded (Lines 119-136). Ref 26-29 have been added.

‘’Even though most MSCs are used for their paracrine signaling effects rather than for expected differentiation or engraftment into host tissue, they still need to persist throughout the inflammatory phase and into the remodeling phase for maximal therapeutic benefit [26]. However, there is evidence that cell-mediated alloimmune responses can limit the persistence of MSCs in vivo [27]. In the study by Berglund et al [26], all horses administered non-stimulated HLA-mismatched allo-MSCs developed de novo antibodies on day 14. These cell therapy induced alloantibodies contributed to the targeted killing of MSCs in vivo. Repeated injections of MSC would result in accelerated rejection of the cells, further limiting its beneficial effects and increasing the potential for adverse events. Injection of donor MSCs into less vascularized tissues like tendons and joints may result in different responses than intradermal or intravascular administration, during injury even tendons and joints are infiltrated with immune cells [28] that can contribute to allorecognition and subsequent immune rejection of MSCs. As the therapeutic benefits of MSCs appear to be largely due to the secretion of paracrine factors that promote healing of healthy tissue [29], it is necessary for the cells to persist throughout the initial inflammatory and healing period. Targeted destruction of allogeneic, MHC mismatched MSCs shortly after transplant would therefore limit their therapeutic potential.  ‘’

Point 2. A close attention should be payed when describing data on alloantibodies. Some studies reported the level of panel reactive antibodies (cPRA) which are not necessarily donor-specific (for example, refs 39 and 44). Similarly, presence of anti-HLA antibodies (not always the HLA expressed by allogeneic MSC) before the injection of allogeneic MSC could indicate pre-sensitization toward allogeneic HLA. This situation should be discussed separately from de-novo DSA development against MSC, since pre-formed antibodies can destroy the infused cells, while de-novo generation of MSC donor-specific antibodies can eliminate these cells in case of repeated injections. In addition, de-novo generation of MSC donor-specific antibodies can expose the patient to some risks, such as the low accessibility to a future organ transplantation.

Studies that measure the development of alloantibodies with cPRA and those that measure DSAs have been differentiated in the text.

Lines 227-229:’’ In the review the alloantibody data are organized into two groups (Table 3). On the one hand, seven studies measured alloantibodies with cPRA (calculated- Panel Reactive Antibodies) [46–54]. And 6 studies measured the presence of DSA [55–59].’’

In addition, a column has been added to Table 3 specifying the type of measurement that was carried out in each study (see Table 3). To differentiate in the text those patients who were positive in cPRA from those who generated DSA, the percentages have been calculated separately (See abstract and Results section).

Besides, the entire text has been modified to make it more understable. When referring to studies that measured DSA, ‘’DSA’’ appears in the text; when referring to studies carried out with cPRA, ‘’cPRA’’ appears in the text; and when referring to several studies and each one performed a type of measurement, ‘’alloantibodies’’ appears in the text. To differentiate in the text those patients who were positive in cPRA from those who generated DSA, the percentages have been calculated separately (See abstract and Results section).

On the other hand, in the discussion section, (Lines 447-469) the presensitization status of the patients and the de-novo generation of DSA have been discussed separately (both appear in two separate questions).

Point 3. Ref 43 and 46 did not report any data on DSA development, please address this inconsistency.

Ref 43 and 46 (now ref 48 and 51) refer to two clinical trials, specifically NCT01586312 and NCT01860417. In the referenced articles [48,51] the basic information of the clinical trials is extracted to carry out the systematic review. The DSA development data was extracted from an article referring to both clinical trials (we included the reference 50 (Garcia-Sancho et al. 2017).

Point 4. In ref 50 only a subgroup of patients given allo-MSC had anti-HLA antibodies measured. Please re-calculate and provide the exact number of patients analyzed overall.

Taking into account that in this study the development of DSA was only measured in a subgroup of patients, the total number of patients included in the analysis changes to 555, and the percentage of sensitized patients changes to 11.51 %. The data has been corrected in the abstract, tables 1 and 2, and in the sections ‘’ Material and methods ’’ ‘’ Results ’’and ‘’ Discussion ’’.

Point 5. Line 128-130. The sentence appears to be incorrect. The generation of DSA is likely the results of indirect recognition of MSC HLA presented by patient APC to CD4+ T cells.

The sentence in Lines 155-156 has been rewritten.

Round 2

Reviewer 2 Report

The Authors have addressed appropriately my previous concerns and I wish to thanks them for their effort.

I have only an additional minor suggestion.

For consistency I would remove the term “efficacy” also from lines 159, 163, 308 and 541.

Author Response

We would like to thank you for taking the time and effort necessary to review our manuscript.

We sincerely appreciate all valuable comments and suggestions, wich helped us to improve the quality of our review.

The minor suggestions have been applied in the manuscript, so that we have eliminated the term “efficacy” also from lines 159, 163, 308 and 541.
